**Data Availability Statement:** All files are available from OSF: https://osf.io/3h5a7.

**Funding:** The authors received no specific funding for this work.

# Personal and psychosocial factors of burnout: A survey within the French neurosurgical community

**Clément Baumgarten**[1]*, **Estelle Michinov**[2], **Géraldine Rouxel**[2], **Vincent Bonneterre**[3], **Emmanuel Gay**[1], **Pierre-Hugues Roche**[4]

1 Department of Neurosurgery, University Hospital of Grenoble, Grenoble, France, 2 Psychology Laboratory, Cognition, Behavior, Communication, University of Rennes 2, Rennes, France, 3 Department of Occupational Medicine, University Hospital of Grenoble, Grenoble, France, 4 Department of Neurosurgery, University Hospital of Marseille, Marseille, France

* cbaumgarten@chu-grenoble.frn

## Abstract

### Object

The neurosurgical community is particularly exposed to burnout. The objectives of this study were to report the prevalence and associated factors of burnout within the French neurosurgical community using validated academic and psychologic scales.

### Methods

A national survey was sent to 141 French residents and 432 neurosurgeons between April and July 2019. Burnout was surveyed using the Maslach burnout inventory. The survey included demographic data and several academic psychologic scales. A stepwise multiple regression was used to determine factors that are associated with burnout scores.

### Results

The response rate was 100% and 23.6% for residents and neurosurgeons, respectively. Prevalence of burnout within the French neurosurgical community was 49%. There were no significant differences between residents and neurosurgeons. Two categories of factors were associated with the main dimensions of burnout during the stepwise multiple regression: personality and factors related with neurosurgical practice. Personality types such as neuroticism were negatively associated with burnout while agreeableness was protective. Work addictive profile with excessive work and absorption at work were negatively associated. Factors associated with neurosurgical practice such as conflict of work into family life, unbalanced effort to reward ratio, work duration were negatively associated. Pleasure at work was protective.

### Conclusion

Prevalence of burnout is high among French neurosurgeons. Predictive models can be used to identify and prevent burnout among profiles at risk.

**Competing interests:** The authors have declared that no competing interests exist.

## Introduction

Workers in many occupational sectors are susceptible to creating a situational context that leads to burnout symptoms [1]. Burnout is a syndrome defined by emotional exhaustion (EE), feelings of depersonalization (DP) and a lack of personal accomplishment (PA) in relation to professional activity [2]. Studies have demonstrated that chronic stressors could increase burnout arise from an imbalance between job demands and job resources [3]. Burnout is associated with several comorbid factors such as chronic fatigue, addictive behaviors, substance use and suicidal ideation, and different health complains [4]. Studies suggest that burnout has a negative impact on psychological and physical health of workers, but also on their interpersonal relationships and job environment. For healthcare professionals and physicians, burnout affects the quality of patients' care through increased medical errors, decreased empathy and decreased productivity at work [5]. Thus, efforts to identify and prevent burnout should lead to better health at work and better quality of care. The prevalence of burnout within different surgical specialties is about 40% [6], and for certain subspecialties such as neurosurgery, the prevalence of burnout is estimated to be 27% to 56.7% [7–9]. Neurosurgery is a demanding specialty in medicine. Work hours are among the longest [10] as the number of night shifts, and has a heavy medicolegal burden due to high malpractice risk [11].

In different American studies [7,8], prevalence and associated factors of burnout in residents and neurosurgeons have been reported. Surprisingly, Shakir et al. [7] reported a burnout rate of 36.5%, Attenello et al. [8] a 67% rate for residents, with an association with inadequate operative exposure, and social stressors while mentorship was a protective factor. McAbee et al. [9] reported for board certified neurosurgeons a 56.7% rate with an association with an unbalanced work and family life and anxiety over future earnings and health care reform. Still, burnout prevalence may differ depending on the health care system and further studies are needed to understand such a complex phenomenon. Moreover, psychological profiles and validated work-related academic scales have never been used when studying burnout in neurosurgery.

In this paper, we present the results of a nationwide survey targeting the entire neurosurgical community in France: residents and neurosurgeons, including those in private practice. The objectives of this work were to report the prevalence and associated factors of burnout within the French neurosurgical community using validated academic and psychologic scales.

## Material and methods

### Participants

At the time of the survey there were 587 board certified neurosurgeons. 354 worked in a public hospital, 139 were exclusively in private practice and 94 worked in both. There were 141 residents in the accredited French residency program. Contact information was obtained from the French Society of Neurosurgery. 141 and 432 email addresses were available for residents and neurosurgeons, respectively. An email with an anonymized link to the study survey was sent to the population mentioned above. For residents, the survey was part of the national days of neurosurgery teaching. The content of the mail is provided in the supplementary material (S1 Appendix). The mail was sent in April 2019 and two follow up requests were sent in May and June 2019. Responses were collected in July 2019. Survey responses were anonymous. Completing all of the questions was mandatory, so there was no missing data. Written consent was not required due to the non-mandatory nature of this questionnaire. The questionnaire was declared to the CNIL, the French Commission of Informatics and Liberty, (n˚ 2211947 v 0). The questionnaire was approved by the ethics committee (ref. IRB00011687).

## Questionnaire

The questionnaire included several work scales and psychologic tests: the Siegrist effort-reward scale [12]; the French DUWAS scale [13] to explore addiction at work; a two sided work-family conflict scale [14]; the work-related flow inventory [15]; the big five inventory for personality traits [16]; the Maslach Burnout Inventory scale [2] and several demographic data sets.

All definitions were based on the MBI scale [2]. Burnout was defined as having high score for emotional exhaustion and/or depersonalization in the MBI. High EE was defined by an EE score ≥30. High DP was defined by a score higher than 12. Low PA was defined by a score lower than 33.

## Statistical analysis

Demographic information was compiled from a series of descriptive statistics. Overall burnout rate and differences in MBI scores, depending on gender and hierarchical status, were reported. Prior to any analysis, we conducted a Cronbach's alpha analysis to explore the reliability of each questionnaire's items. Item 20 of the overinvestment scale as well as item 35 et 41 of the BFI test were excluded. Cronbach's alphas are reported in S2 Table. Gender and hierarchical status comparison of severe burnout was done using a chi-square test for association. All expected cell frequencies were greater than five. An independent-sample t-test was run to determine if there were differences in MBI scores between genders. An independent-samples test was also run for worked hours and number of night shifts comparison depending on resident and neurosurgeons. A one-way ANOVA was run to determine if there were differences in MBI score between professional status (i.e. fellows, professor. . .). Prior assumptions were checked when one-way ANOVA was used for mean comparisons between multiples groups. There were no outliers, as assessed by boxplot; data was normally distributed for each group, as assessed by Shapiro-Wilk test (p > .05); and there was homogeneity of variances, as assessed by Levene's test of homogeneity of variances (p = >0.05). A Tukey post hoc test was used when the result of the one-way ANOVA showed significance.

We conducted a stepwise multiple regression to determine factors that are associated with burnout scores. The list of entered factors is available in S2 Table. There was linearity as assessed by partial regression plots and a plot of studentized residuals against the predicted values. There was independence of residuals, as assessed by a Durbin-Watson statistic of respectively 2.02, 1.97, 2.045 for EE, DP and PA scores. There was homoscedasticity, as assessed by visual inspection of a plot of studentized residuals versus unstandardized predicted values. There was no evidence of multicollinearity, as assessed by tolerance values greater than 0.1. There were no studentized deleted residuals greater than ±3 standard deviations, no leverage values greater than 0.2, and values for Cook's distance above 1. There assumption of normality was met, as assessed by Q-Q Plot. All data were analyzed using the SPSS 24.0 statistical software (IBM, Armonk, New York).

## Results

141 residents (100%) completed the survey as it was part of national days of teaching of neurosurgery in France. Of the 432 contacted neurosurgeons, 102 (23.6%) completed the survey. The demographic characteristics of the respondents are shown in Table 1. Thus, more than half of the population were residents (57.3%). In the overall population: 26.6% were women; 28% of the residents and 22% of the neurosurgeons.

Fig 1 represents the mean declared working time per week and number of night shifts of residents and neurosurgeons. There was a significant difference depending on hierarchical status in working time: mean working time n = 69.65h and n = 57.08h for residents and

**Table 1. Demographic data.**

| Sex | n | % |
|---|---|---|
| Men | 179 | 73.4 |
| Women | 64 | 26.6 |
| Professional status | | |
| Resident | 141 | 58 |
| 1$^{st}$ year | 26 | 10.6 |
| 2$^{nd}$ year | 24 | 9.8 |
| 3$^{rd}$ year | 23 | 9.3 |
| 4$^{th}$ year | 31 | 12.6 |
| 5$^{th}$ year | 37 | 15.04 |
| Fellow | 22 | 9.1 |
| Attending neurosurgeon | 31 | 12.8 |
| Professor | 21 | 8.6 |
| Private practice | 28 | 11.5 |
| Children | | |
| Yes | 92 | 37.9 |
| No | 151 | 62.1 |
| Marital status | | |
| Single | 79 | 32.5 |
| Common law | 72 | 29.6 |
| Married | 84 | 34.6 |
| Divorced | 8 | 3.3 |
| Security rest | | |
| Yes | 77 | 30.9 |
| No | 90 | 36.6 |
| It depends | 79 | 32.5 |

neurosurgeons, respectively (p<0.001). There was also a significant difference for the number of shifts with residents declaring the most shifts: n = 5.91 and n = 3.76 for residents and neurosurgeons, respectively (p<0.001).

Table 2 reports an overall burnout rate of 49% within the French neurosurgical community. This is a prevalence and reflects the burnout rate at the time of the study i.e. July 2019. A comparison of MBI scores is reported depending on gender and hierarchical status. There was no significant difference between gender and hierarchical status for severe burnout. There was only a significant difference between residents and neurosurgeons for DP (p = 0.002). There was a significant difference depending on hierarchical status for depersonalization according to the one-way ANOVA test. However, there were conflicting results with a Tukey post hoc test that did not show significant differences between groups.

Table 3 reports the stepwise multiple regression analysis demonstrating the associations between burnout categories and several factors. Two types of variables are associated with burnout categories. Personality dimensions appear to be an independent variable with neurosurgical practice. Neuroticism was associated with EE. Agreeableness was a protective factor against depersonalization and has a positive impact on personal accomplishment. Openness to experience, extraversion and consciousness was associated with personal accomplishment. Fig 2 represents the mean BFI scores for each dimension within the French neurosurgical community. Neurosurgeons appear as introverted, conscientious, compassionate, inventive, and rather secure regarding the neuroticism dimension. There were variables related to neurosurgical practice: the conflict of work on family life has a negative impact on emotional exhaustion

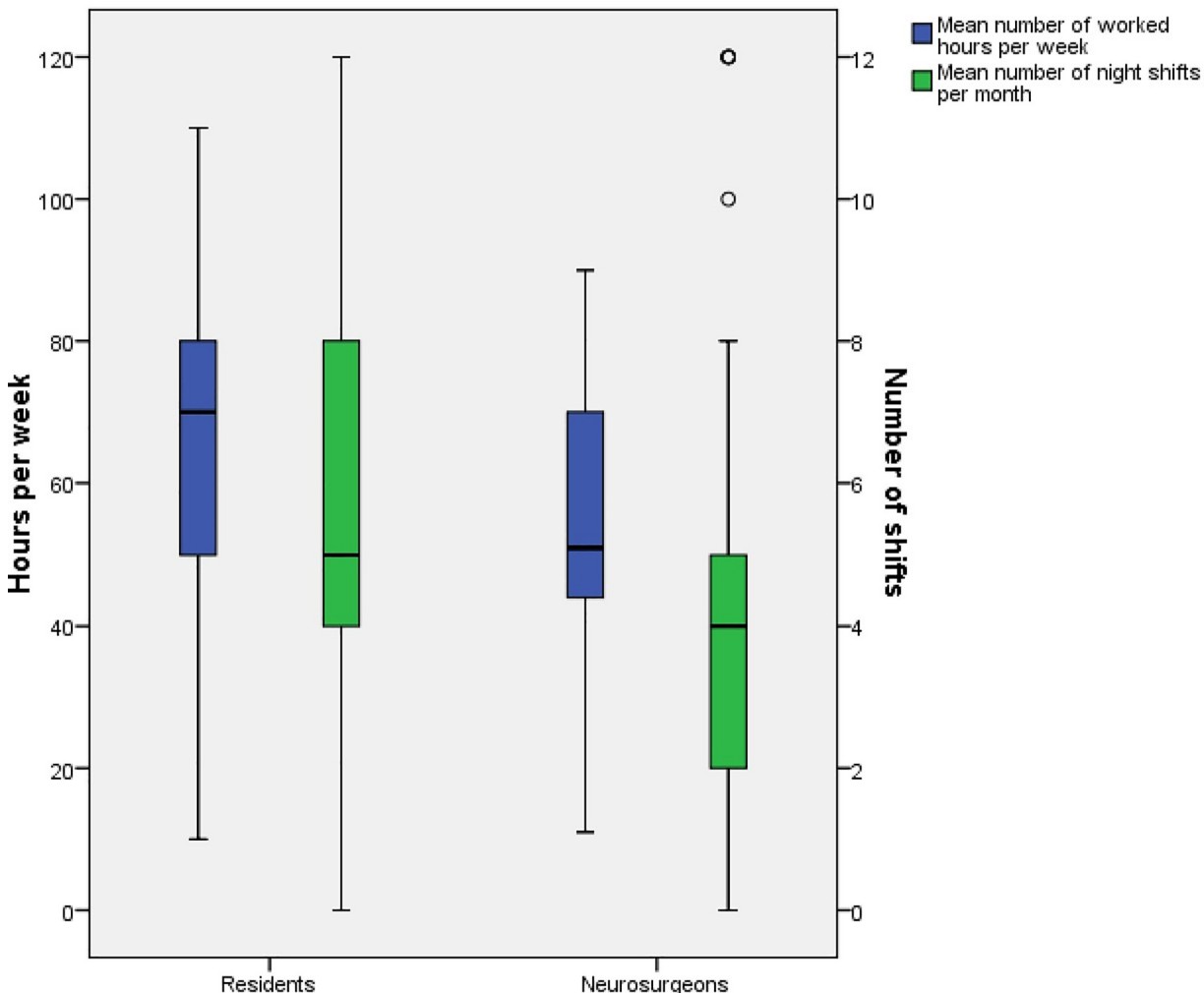

**Fig 1. Worked hours per week and number of night shifts per month depending on hierarchical status.**

and depersonalization. Pleasure at work was negatively associated with EE and DP while it was positively associated with PA. Absorption at work has a negative association with EE. An unbalance effort/reward ratio has a negative association with EE but positive association with PA. Excessive work according to Duwas scale was associated with EE. Rise on hierarchical status has a positive association with DP. Finally, working hours have a negative association with EE. The univariates analysis and direction of each associated factors obtained during the stepwise multiple regression are represented in the S1 Fig.

Table 4 reports the suicidal thoughts and drug uses within the French neurosurgical community. Use of drugs and consultation of a psychiatrist were significantly associated with burnout. There was only a non-significant trend with suicidal thoughts.

## Discussion

This study assessed burnout within the French neurosurgical community and explored associated factors. To our knowledge, this is the first study that assessed burnout rate of neurosurgeons and residents in neurosurgery at the same time and with the same variables, giving a full picture of a community. Overall burnout rate was 49% and no significant differences between

**Table 2. Overall burnout rate and differences in MBI scores depending on gender and hierarchical status.**

|  |  | N | Emotional exhaustion | | Personal accomplishment | | Depersonalization | | Burnout Proportion |
|---|---|---|---|---|---|---|---|---|---|
|  |  |  | Mean | (SD) | Mean | (SD) | Mean | (SD) |  |
| **Overall** |  | 243 | 23.84 | 11.01 | 33.04 | 8.2 | 10.41 | 6.21 | 49% |
| **Gender** |  |  |  |  |  |  |  |  |  |
| Women |  | 64 | 23.79 | 10.94 | 33.5 | 8.34 | 9.3 | 5.95 | 48.4% |
| Men |  | 179 | 23.84 | 10.07 | 32.88 | 8.16 | 10.79 | 6.28 | 49.2% |
| p-value |  |  | 0.97 |  | 0.61 |  | 0.12 |  | 0.92 |
| **Professional status** |  |  |  |  |  |  |  |  |  |
| Residents |  | 141 | 23.22 | 11.14 | 32.43 | 9.03 | 11.45 | 6.45 | 52.5% |
| Neurosurgeons |  | 102 | 24.68 | 10.82 | 33.90 | 6.82 | 8.99 | 5.56 | 47.5% |
| p-value |  |  | 0.31 |  | 0.32 |  | **0.002** |  | 0.24 |
|  | Fellows | 22 | 25.78 | 10.47 | 33.27 | 4.89 | 10.04 | 6.34 | 54.5% |
|  | Senior | 31 | 26.81 | 11.57 | 32.48 | 7.53 | 8.09 | 5.56 | 48.4% |
|  | Professor | 21 | 24.14 | 10.24 | 36.24 | 5.81 | 7.81 | 4.64 | 38.1% |
|  | Private practice | 28 | 21.89 | 10.59 | 34.21 | 7.78 | 9.14 | 5.75 | 35.7% |
|  | p-value |  | 0.37 |  | 0.32 |  | **0.03** |  | 0.51 |

hierarchical status were found. Two types of variables were found: 1) the impact of personality as an independent factor with neurosurgical practice and 2) factors associated with neurosurgical practice measured with validated academic scales. In the following sections, we discuss

**Table 3. Stepwise multiple regression analyses to test the relative contributions of different predictors on burnout dimensions.**

|  | β | t | p | $R^2$ | F | p |
|---|---|---|---|---|---|---|
| **Emotional exhaustion** |  |  |  | .524 | 36.47 | **< .001** |
| Big Five Neuroticism dimension | .22 | 4.46 | < .001 |  |  |  |
| Work to family conflict | .31 | 6.02 | < .001 |  |  |  |
| Effort/Reward Ratio | .18 | 3.72 | < .001 |  |  |  |
| Pleasure at Work—Flow | -.30 | -5.14 | < .001 |  |  |  |
| Absorption at Work–Flow | .13 | 2.18 | .03 |  |  |  |
| Excessive Work–Workaholism | .11 | 1.94 | .054 |  |  |  |
| Work duration | .10 | 2.01 | .046 |  |  |  |
| **Depersonalization** |  |  |  | .323 | 22.29 | **< .001** |
| Big Five Agreeableness dimension | -.18 | -3.17 | .002 |  |  |  |
| Work to family conflict | .25 | 4.13 | < .001 |  |  |  |
| Overinvestissment | .16 | 2.69 | .008 |  |  |  |
| Pleasure at Work—Flow | -.25 | -4.36 | < .001 |  |  |  |
| Hierarchical Status | .26 | 4.68 | < .001 |  |  |  |
| **Personal accomplishment** |  |  |  | .339 | 19.89 | **< .001** |
| Big Five Agreeableness dimension | .26 | 4.44 | < .001 |  |  |  |
| Big Five Openness dimension | .19 | 3.19 | .002 |  |  |  |
| Big Five Extraversion dimension | .17 | 2.94 | .004 |  |  |  |
| Big Five Consciousness dimension | .16 | 2.75 | .006 |  |  |  |
| Pleasure at Work—Flow | .19 | 3.31 | .001 |  |  |  |
| Effort/Reward Ratio | .15 | 2.64 | .009 |  |  |  |

Professional Status (coded 1 = Residents; 0 = Clinical practitioners in private and public sectors). Work duration = continuous variable.

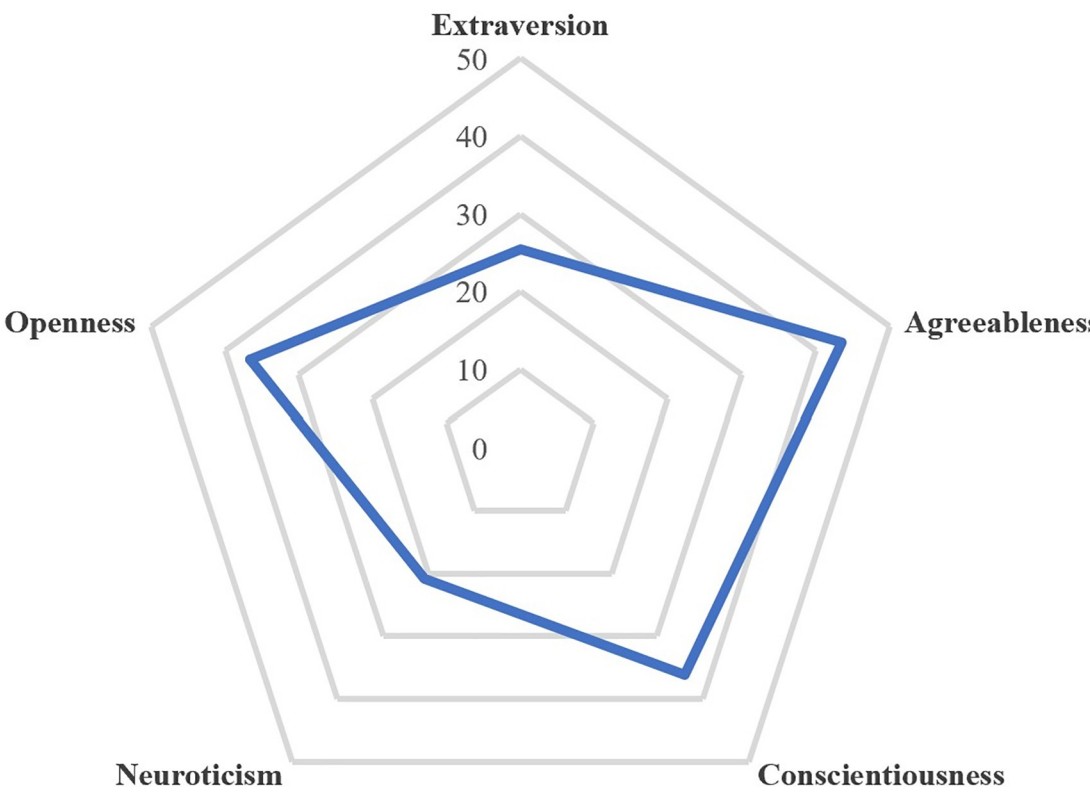

**Fig 2. Big five personality traits mean scores within the French neurosurgical community.**

the results and the proposed multivariate model to predict and prevent burnout in neurosurgery.

## Burnout rate and comparisons

Burnout has been largely described in the medical field [17] and even among surgical communities [18]. Little data exists in neurosurgery event though this is an extreme specialty in terms of working hours [10] and medicolegal burden [11]. American studies showed an overall burnout rate of 56.7% for neurosurgeons [9] and two studies reported either a 36.5% [7] and 67% [8] rate for residents. Keeping the same burnout definition, burnout among Lithuanian neurosurgeons [19] vary between 25.8–41.9%. In this study we reported an overall burnout rate of

**Table 4. Association between suicidal thoughts, drug uses, psychological needs and burnout.**

| Suicidal thoughts | n | % | p-value |
|---|---|---|---|
| Never | 173 | 70.3 | |
| Rarely | 61 | 24.8 | |
| Often | 5 | 12.3 | |
| Yes, with intention to act | 7 | 2.85 | 0.062 |
| Psychiatry, psychology | 22 | 8.9 | **0.019** |
| Antidepressant | 11 | 4.47 | 0.32 |
| Sleeping pill | 21 | 8.5 | 0.42 |
| Drugs | 15 | 6.1 | **0.013** |
| Performance enhancer | 8 | 3.3 | 0.52 |

49% which is higher than the 29% rate in comparison with the most comprehensive and complete meta-analysis of French physicians [17] following the same burnout definition. As for surgeons, Campbell et al. [18] reported a 32% rate in a population of American surgeons.

As the female proportion is growing in neurosurgery [20] and some studies tend to demonstrate that females were more susceptible to burnout [21], we investigated if burnout rate differs between genders. In this study there was no significant difference between genders. There are 28% of women among residents and 22% among neurosurgeons. A growing percentage of women among neurosurgeons should not affect burnout rate in the years to come.

Although there was a trend of decreased burnout rate when one advanced in hierarchical status, no significant difference was shown in the analysis. Lack of power due to lower response rate among neurosurgeons can explain this tendency. There was no difference in the effort/reward ratio of residents and neurosurgeons (0.7). Hierarchical status was only associated with depersonalization in the stepwise multiple regression. The only variables that appear to be statistically different are shifts per month (5.91 vs 3.76: p<0.001) and working hours (69.65 vs 57.08 p<0.001) for residents and neurosurgeons, respectively. Data in the literature finds inadequate operative exposure, lack of opportunities for professional development, and dissatisfaction with colleagues are all common factors between residents and neurosurgeons [8,9]. Residents face problems like working hours, night shifts, social stressors and lack of mentorship [8]. As their careers advance, neurosurgeons face unbalance between work and life outside of the hospital and anxiety over future earnings and/or health care reform [9]. Our data suggests that burnout rate does not seem to vary during the career. From residency to retirement, physicians face different problems with potential negative impacts during the advancement of their career. Zoer et al. [22] drew similar conclusions when they studied associations between psychosocial workload and mental health complaints in different age groups in a railway company. Understanding those subtle differences is crucial to provide accurate interventions.

## Impact of personality

Personality appears as an independent variable within neurosurgical practice. The stepwise multiple regression analysis showed a negative impact of the neuroticism factor for emotional exhaustion. Openness to experience, conscientiousness, extraversion and agreeableness was associated with personal accomplishment. Agreeableness was a protective factor against depersonalization. To our knowledge, this is the first study to explore personality in neurosurgery. Association between personality and burnout has been reported in several work domains [23,24] and also among physicians [25,26]. There are suggestions that surgeons are more likely to be associated with extraversion and openness to experience while pediatricians and family practitioners are more likely to correspond to higher neuroticism [27]. Though neurosurgery is more likely to be exposed to stress and medicolegal burden [11,28], personality traits seems to be protective against burnout in this specialty. In our sample, personality traits tend to show higher levels on agreeableness, openness to experience and conscientiousness and low levels on extraversion and neuroticism. Personality modification does not seem either ethical or desirable. Still, cognitive and psychosocial interventions [29,30] that reduce stress, and anxiety while increasing social support and self-esteem might reduce the complex impact of personality factors on burnout. Also, personality screening at the beginning of residency can be measured to follow personality traits that are more likely to develop burnout. This strategy has been adopted in other fields with high responsibilities and high levels of competence such as special forces in the army [31,32] or airline pilots [33].

### Factors associated with neurosurgical practice

Work to family conflict is a well-documented association with burnout. Jensen et al. [34,35] rigorously demonstrated in a longitudinal study the reciprocal relationships between work to family conflict, emotional exhaustion and psychological health complaints in a population with a lot of business travels. In our study, work to family conflict was the first associated variable in both emotional exhaustion and depersonalization regression models. The results are in line with numerous other studies [36–38]. This is not surprising; neurosurgery is a specialty with night shifts, work hours are among the longest [10], there is some work schedule inflexibility and shiftwork is irregular. All of those factors are associated with work to family conflict [39]. According to the conclusion of Jensen et al, it is likely that today's organization in neurosurgery produces a vicious circle leading to burnout and its consequences. Better organization of continuity of care to clear regular family time may improve this factor.

While high score of absorption according to the WOLF of Bakker et al. [15] has a negative impact, work enjoyment is a protective factor against emotional exhaustion and depersonalization. Neurosurgeons seems to particularly enjoy operative exposure [9] or more broadly clinical work [19] while administrative tasks, which seem to take one-sixth of a physician's time [40], lower career satisfaction. Recruiting administrative assistants whose training time is not as long as that of a neurosurgeon could be a strategy to improve neurosurgeons' clinical productivity as well as improving health at work. Finally, according to Siegrist et al. [12], improvement in rebalancing the effort /reward ratio, which appeared as an independent factor of emotional exhaustion, may reduce burnout.

More modestly, working hours have a negative association. This seems especially true for residents (mean declared working time 69.65h per week) who work at least 20h more than the EU working time directive (48h per week). According to previous discussion, we did not explore the concept of quantity of work weighted by quality. There might be a shift of administrative burden from neurosurgeons to residents that needs to be explored to explain these results. The status of residents seems to expose to depersonalization while advancing during the career seems to protect from depersonalization.

### Associations between burnout, suicidal thoughts and drug uses

Burnout as a syndrome does not have a particular impact but has been described in literature as associated with a negative impact for patient's care and the own physician's health [5,41]. Though it was not the main primary objective of the study we assessed the associations between burnout and health consequences in our studied population. Suicidal thoughts showed a trend that was not statistically significant. Psychiatry or psychologic consultation was associated as well as drug usage. Those results confirm previous findings, [42–45] and actions to prevent burnout may improve physicians' health in neurosurgery.

### Limitations

There are some usual limitations regarding survey studies. There was a 100% answer rate for the residents because it was part of national teaching days. It was a common decision by the French national society of neurosurgery and the college of neurosurgery teachers to make it part of this pedagogic events. There were no significant outliers in the residents' responses after visual inspection of the boxplots. The neurosurgeons were harder to reach with a 23.6% response rate which is rather low in such studies but comparable to the answer rate of other neurosurgical surveys [8,9,46]. This rate was exposed to a selection bias, attending physicians are usually less involved in academic work and are under-represented in this study.

This was not the first study about burnout in neurosurgery. We chose to explore different variables associated with burnout to further understand the phenomenon. Previous studies identified mainly dichotomous variables such as operative exposure and binary feeling of unbalance work and family life [7–9,19]. In this study we used quantified validated scales that help to understand and explain previous results.

Survey studies cannot be exhaustive, and a longer questionnaire would have lowered the response rate. Neurosurgeons shares many well documented associated factors of burnout with other occupational groups. While we did not measure occupational stress in this study it is certainly a shared features, notably with lawyers [47] and non-medical occupational health staff [48]. Further studies are required for further comprehension of this complex phenomenon. Moreover, as it is a transversal study, we can only identify associations related to burnout. Further studies will address the causality between those variables and burnout.

## Conclusion

We have reported the prevalence and factors associated with burnout within the French neurosurgical community from the residents to the professors. Burnout prevalence is high in comparison with physicians and even among surgeons. Two categories of factors were identified. Personality types have an independent association regarding neurosurgical practice. Still, inherent factors associated with neurosurgical practice such as conflict of work into family life, unbalanced effort/reward ratio and work duration exist. Thus, this study may identify two levers of action to prevent and reduce burnout. A preliminary screening of personality may be proposed for a closer follow-up during residency. Improvement of working conditions, division of tasks of continuity of care and a readjustment of the effort / reward ratio may lower burnout in neurosurgery.

A prospective follow-up of the French neurosurgical cohort may answer if those levers of action can be effective to improve health at work in neurosurgery.

## Supporting information

**S1 Appendix. Questionnaire.**
(PDF)

**S1 Table. Cronbach alphas.** Alphas on the diagonal: [a] after elimination of item 20; [b]: after elimination of items 35 and 41.
(DOCX)

**S2 Table. Variables entered during the stepwise multiple regression.**
(DOCX)

**S1 Fig. Univariate analysis of each associated factors with the main dimensions of burnout.**
Factors associated with emotional exhaustion: a, b, c, d, e, f, g. a: linear regression between emotional exhaustion and pleasure at work; b: linear regression between emotional exhaustion and neuroticism; b: linear regression between emotional exhaustion and work to family conflict; c: linear regression between emotional exhaustion and effort/reward ratio; d: linear regression between emotional exhaustion and pleasure at work; e: linear regression between emotional exhaustion and absorption at work; f: linear regression between emotional exhaustion and excessive work; g: linear regression between emotional exhaustion and work duration. Factors associated with depersonalization: h, i, j, k, l. h: linear regression between depersonalization and agreeableness; i: linear regression between depersonalization and work to family conflict; j: linear regression between depersonalization and overinvestment; k: linear regression between depersonalization and pleasure at work; l: linear regression between

depersonalization and hierarchical status. Factors associated with personal accomplishment: m, n, o, p, q, r. m: linear regression between personal accomplishment and agreeableness; n: linear regression between personal accomplishment and openness; o: linear regression between personal accomplishment and extraversion; p: linear regression between personal accomplishment and consciousness; q: linear regression between personal accomplishment and pleasure at work; r: linear regression between personal accomplishment and effort/reward ratio.
(TIFF)

## Acknowledgments

We would like to thank Blake Fleck for revising the English style.

Portions of this work has been presented (oral communication, plenary) during the annual research meeting of the French National Academy, Paris, France, and has been awarded the "Risk Prevention Prize", the 22nd of November 2019.

## Author Contributions

**Conceptualization:** Clément Baumgarten, Estelle Michinov, Pierre-Hugues Roche.

**Data curation:** Clément Baumgarten.

**Formal analysis:** Clément Baumgarten, Estelle Michinov, Géraldine Rouxel.

**Investigation:** Clément Baumgarten.

**Methodology:** Clément Baumgarten, Estelle Michinov, Géraldine Rouxel.

**Supervision:** Emmanuel Gay, Pierre-Hugues Roche.

**Writing – original draft:** Clément Baumgarten.

**Writing – review & editing:** Clément Baumgarten, Estelle Michinov, Géraldine Rouxel, Vincent Bonneterre, Emmanuel Gay, Pierre-Hugues Roche.

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
