## [Decision Letter · Decision Letter 0]

27 Feb 2020

PONE-D-20-00752

Personal and Psychosocial factors of Burnout: A Survey within the French Neurosurgical Community.

PLOS ONE

Dear MR Baumgarten,

Thank you for submitting your manuscript to PLOS ONE. After careful consideration, we feel that it has merit but does not fully meet PLOS ONE’s publication criteria as it currently stands. Therefore, we invite you to submit a revised version of the manuscript that addresses the points raised during the review process.

Our reviewers have identified a certain potential in your submitted manuscript, highlighting its possible contribution to the development of applied measures and strategies for the improvement of the job quality and physical/mental health of specialized healthcare providers. Even though, you will find a relatively short (but substantial) set of comments and suggestions that you should consider responding to, in order to improve the manuscript and its scientific value.

Also, and as a personal suggestion, please consider updating the theoretical framework of the paper, taking into account recent studies on (e.g.) burnout and health issues among vulnerable occupational groups apart from physicians and healthcare providers, that share key factors such as a highly stressful work environment, time pressure and limited degrees of skill discretion. For instance, the relationship between EE (Emotional Exhaustion) and different health complaints affecting working populations (same as among medical personnel) is a factor that recent sources strongly highlight, and may enrich the comparability of your findings. This might contribute to strengthen the explanatory potential of the findings and empirical support given to your discussion.

We would appreciate receiving your revised manuscript by Apr 12 2020 11:59PM. To enhance the reproducibility of your results, we recommend that if applicable you deposit your laboratory protocols in protocols.io, where a protocol can be assigned its own identifier (DOI) such that it can be cited independently in the future. For instructions see: http://journals.plos.org/plosone/s/submission-guidelines#loc-laboratory-protocols

We look forward to receiving your revised manuscript.

Kind regards,

Sergio A. Useche, Ph.D.

Academic Editor

PLOS ONE

Journal Requirements:

Reviewers' comments:

Reviewer's Responses to Questions

**Comments to the Author**

1. Is the manuscript technically sound, and do the data support the conclusions?

Reviewer #1: Yes

Reviewer #2: Partly

2. Has the statistical analysis been performed appropriately and rigorously? 

Reviewer #1: Yes

Reviewer #2: Yes

3. Have the authors made all data underlying the findings in their manuscript fully available?

Reviewer #1: Yes

Reviewer #2: Yes

4. Is the manuscript presented in an intelligible fashion and written in standard English?

Reviewer #1: Yes

Reviewer #2: Yes

5. Review Comments to the Author

Reviewer #1: Considerations:

The object of the study is very significant. Burnout is a syndrome that appears in several professional categories and affects the care of the population, due to causing mental exhaustion in the affected professionals.

1- The search objective is not clearly defined in the Abstract.

2- Methods should not contain the objectives (line 59). The objectives should be described in the previous session.

Reviewer #2: This ms reports results of a well-designed study that used validated assessment instruments to evaluate the role of personality, effort/reward ratio, and other factors like pleasure at work-flow to identify personal and psychosocial factors associated with burnout in the French neurosurgical community. There are some concerns the authors need to address to maximize the impact of this study.

1. While the associations documented by the rigorous statistical analyses are strong, it will help the reader understand the nature of those associations if the authors provided figures (or tables) that illustrate the directions of those associations – e.g., graphs showing levels of Emotional Exhaustion as Neuroticism increases, Depersonalization as Agreeable decreases, Personal Accomplishment as Effort/Reward ratio changes.

2. While they are probably correct in noting that “personality modification does not seem either ethical or desirable,” it could help appreciate the implications of the associations they have found if they consider the potential paths whereby the personality dimensions lead to burnout – e.g., high neuroticism � high depression �high emotional exhaustion. Interventions that reduce levels of such mediators could reduce the impact of personality factors on burnout. In a study of Chinese medical students, for example, Chun et al. (Efficacy of Williams LifeSkills training for improving psychological health: a pilot comparison study of Chinese medical students. Asia Pac Psychiatry. 2014;6(2):161-9. PMID: 23857943) found that training in cognitive behavioral stress management and interpersonal interaction skills produced improvements in anxiety, depression, negative coping, social support, and self-esteem.

3. Some minor concerns need to be addressed:

a. line 82 – “Maslach Burnout Inventory scale6” – what does the 6 refer to?

b. line 133 – The only significant difference between residents and neurosurgeons was for DP in Table 2, not PA as stated in the text.

c. line 151 – They say Pleasure at work is positively associated with each burnout dimension, but inspection of Table 3 show that Pleasure at Work-Flow is negatively associated with EE and DP, but positively associated with PA – suggesting that high Pleasure at Work is associated with lower EE and DP but higher PA, all of which makes sense. As per comment 1 above, it would help the reader understand these associations if figures were provided to illustrate them.

6. PLOS authors have the option to publish the peer review history of their article (what does this mean?). If published, this will include your full peer review and any attached files.

Reviewer #1: No

Reviewer #2: No

---

## [Author Response · Author response to Decision Letter 0]

20 Mar 2020

Editors

Plos ONE

Ref : PONE-D-20-00752R1

Personal and Psychosocial factors of Burnout: A Survey within the French Neurosurgical Community

To Sergio A. Useche, Academic Editor

Dear Sir,

Please find enclosed a revised manuscript entitled “Personal and Psychosocial factors of Burnout: A Survey within the French Neurosurgical Community” by Clément Baumgarten, Estelle Michinov, Géraldine Rouxel, Vincent Bonneterre, Emmanuel Gay and Pierre-Hugues Roche. This manuscript is submitted for consideration after revisions for publication in the PLOS ONE journal. We have addressed and clarified all the concerns that were raised by the academic editor and the reviewers. We believe that this revised version results in a much stronger paper for your journal. All the authors have reviewed and approved this revised version of the manuscript. 

In addition to the revised version of our manuscript (in Word format), please find enclosed the following documents:

- A point-by-point list of all changes made in response to suggestions of the reviewers.

- A version of the manuscript in “track changes” format, with changes highlighted (in Word format).

In the hope that you find that this paper falls within the scope of PLOS ONE, and that it will be of interest to your readers, I look forward to hearing from you soon. Please do not hesitate to contact me if you have any questions.

Yours faithfully, 

Clément Baumgarten

 

Response to reviewers

Ref: PONE-D-20-00752R1

Personal and Psychosocial factors of Burnout: A Survey within the French Neurosurgical Community

I. INTRODUCTION

The authors would like to thank the reviewers for their relevant remarks and suggestions for improving the paper. In this document, we provide a point-by-point response to each of the issues raised. Each issue is written in bold text, the corresponding response is in normal text, and we provide in italics the text added/modified in the paper to address the issue. In the main paper, all changes are clearly marked in yellow.

Each modification in the text of the manuscript was highlighted in red to be more easily visible for the reviewers.

Here is a point-by-point list of all changes made in response to the suggestions of the reviewers:

II. ACADEMIC EDITOR

Also, and as a personal suggestion, please consider updating the theoretical framework of the paper, taking into account recent studies on (e.g.) burnout and health issues among vulnerable occupational groups apart from physicians and healthcare providers, that share key factors such as a highly stressful work environment, time pressure and limited degrees of skill discretion. For instance, the relationship between EE (Emotional Exhaustion) and different health complaints affecting working populations (same as among medical personnel) is a factor that recent sources strongly highlight and may enrich the comparability of your findings. This might contribute to strengthen the explanatory potential of the findings and empirical support given to your discussion.

We thank you for your suggestion to enhance the theoretical discussion of the paper. We revised the introduction accordingly. We added several references to discuss and strengthen our findings. 

Added text: 

Line 45: “Workers in many occupational sectors are susceptible to creating a situational context that leads to burnout symptoms[1] . Burnout is a syndrome defined by emotional exhaustion (EE), feelings of depersonalization (DP) and a lack of personal accomplishment (PA) in relation to professional activity[2]. Studies have demonstrated that chronic stressors could increase burnout arise from an imbalance between job demands and job resources[3]. Burnout is associated with several comorbid factors such as chronic fatigue, addictive behaviors, substance use and suicidal ideation, and different health complains[4]. Studies suggest that burnout has a negative impact on psychological and physical health of workers, but also on their interpersonal relationships and job environment. For healthcare professionals and physicians, burnout affects the quality of patients’ care through increased medical errors, decreased empathy and decreased productivity at work[5]. Thus, efforts to identify and prevent burnout should lead to better health at work and better quality of care. The prevalence of burnout within different surgical specialties is about 40%[6], and for certain subspecialties such as neurosurgery, the prevalence of burnout is estimated to be 27% to 56.7%[7–9]. Neurosurgery is a demanding specialty in medicine. Work hours are among the longest[10] as the number of night shifts, and has a heavy medicolegal burden due to high malpractice risk[11].”

Line 223 : “Zoer et al[18] drew similar conclusions when they studied associations between psychosocial workload and mental health complaints in different age groups in a railway company. Understanding those subtle differences is crucial to provide accurate interventions.”

Line 238: “Work to family conflict is a well-documented association with burnout. Jensen et al[30,31] rigorously demonstrated in a longitudinal study the reciprocal relationships between work to family conflict, emotional exhaustion and psychological health complaints in a population with a lot of business travels.”

Line 255:” According to the conclusion of Jensen et al, it is likely that today’s organization in neurosurgery produces a vicious circle leading to burnout and its consequences.”

Line 299: “Neurosurgeons shares many well documented associated factors of burnout with other occupational groups. While we did not measure occupational stress in this study it is certainly a shared features, notably with lawyers[43] and non-medical occupational health staff[44].”

III. REVIEWER 1

The object of the study is very significant. Burnout is a syndrome that appears in several professional categories and affects the care of the population, due to causing mental exhaustion in the affected professionals.

1- The search objective is not clearly defined in the Abstract. Methods should not contain the objectives (line 59). The objectives should be described in the previous session.

We thank the reviewer 1 for helping us clarify the structure and the objectives of our manuscript. We clarify the objective section accordingly.

Added text line 25: The objectives of this study were to report the prevalence and associated factors of burnout within the French neurosurgical community using validated academic and psychologic scales.

Added text: line 70: “The objectives of this work were to report the prevalence and associated factors of burnout within the French neurosurgical community using validated academic and psychologic scales.”

IV. REVIEWER 2

This ms reports results of a well-designed study that used validated assessment instruments to evaluate the role of personality, effort/reward ratio, and other factors like pleasure at work-flow to identify personal and psychosocial factors associated with burnout in the French neurosurgical community. There are some concerns the authors need to address to maximize the impact of this study.

1. While the associations documented by the rigorous statistical analyses are strong, it will help the reader understand the nature of those associations if the authors provided figures (or tables) that illustrate the directions of those associations – e.g., graphs showing levels of Emotional Exhaustion as Neuroticism increases, Depersonalization as Agreeable decreases, Personal Accomplishment as Effort/Reward ratio changes.

We totally agree and thank the reviewer for this relevant suggestion. We provided a supplemental figure with each univariate analysis that were significant during the stepwise multiple regression. The figure provides the reader the direction of the association.

Added material: S4 Figure. Univariate analysis of each associated factors with the main dimensions of burnout.

Added text: line 172: “The univariates analysis and direction of each associated factors obtained during the stepwise multiple regression are represented in the S4 Appendix.”

2. While they are probably correct in noting that “personality modification does not seem either ethical or desirable,” it could help appreciate the implications of the associations they have found if they consider the potential paths whereby the personality dimensions lead to burnout – e.g., high neuroticism � high depression �high emotional exhaustion. Interventions that reduce levels of such mediators could reduce the impact of personality factors on burnout. In a study of Chinese medical students, for example, Chun et al. (Efficacy of Williams LifeSkills training for improving psychological health: a pilot comparison study of Chinese medical students. Asia Pac Psychiatry. 2014;6(2):161-9. PMID: 23857943) found that training in cognitive behavioral stress management and interpersonal interaction skills produced improvements in anxiety, depression, negative coping, social support, and self-esteem.

Burnout is a complex phenomenon intricated with a multitude of human factors. We agree that those kinds of interventions might affect the factors associated with personality and burnout.

Added text: line 241 : “Still, cognitive and psychosocial interventions[24,25] that reduce stress, and anxiety while increasing social support and self-esteem might reduce the complex impact of personality factors on burnout.”

3. Some minor concerns need to be addressed:

a. line 82 – “Maslach Burnout Inventory scale6” – what does the 6 refer to?

This typography error was corrected.

b. line 133 – The only significant difference between residents and neurosurgeons was for DP in Table 2, not PA as stated in the text.

We thank the reviewer 2 for correcting this mistype error. 

Corrected text line 149: “There was only a significant difference between residents and neurosurgeons for DP”

c. line 151 – They say Pleasure at work is positively associated with each burnout dimension, but inspection of Table 3 show that Pleasure at Work-Flow is negatively associated with EE and DP, but positively associated with PA – suggesting that high Pleasure at Work is associated with lower EE and DP but higher PA, all of which makes sense. As per comment 1 above, it would help the reader understand these associations if figures were provided to illustrate them.

We apologize for this misinterpretation error. We addressed this issue in the first comment answer.

Added text line 167: Pleasure at work was negatively associated with EE and DP while it was positively associated with PA.

---

## [Decision Letter · Decision Letter 1]

17 Apr 2020

PONE-D-20-00752R1

Personal and Psychosocial factors of Burnout: A Survey within the French Neurosurgical Community

PLOS ONE

Dear MR Baumgarten,

Thank you for submitting your manuscript to PLOS ONE. After careful consideration, we feel that it has merit but does not fully meet PLOS ONE’s publication criteria as it currently stands. Therefore, we invite you to submit a revised version of the manuscript that addresses the points raised during the review process.

Please refer to the comments appended below, where one of the reviewers requires you to perform a relatively small (but important) amendment in regard to (i) the reported burnout levels in this group of professionals, and (ii) the scale indicating whether burnout levels are high or low.

Once you send you revisions, and if the quality of the amendments is good enough, I will proceed to accept the paper without a new round of reviews.

We would appreciate receiving your revised manuscript by Jun 01 2020 11:59PM. To enhance the reproducibility of your results, we recommend that if applicable you deposit your laboratory protocols in protocols.io, where a protocol can be assigned its own identifier (DOI) such that it can be cited independently in the future. For instructions see: http://journals.plos.org/plosone/s/submission-guidelines#loc-laboratory-protocols

We look forward to receiving your revised manuscript.

Kind regards,

Sergio A. Useche, Ph.D.

Academic Editor

PLOS ONE

Reviewers' comments:

Reviewer's Responses to Questions

**Comments to the Author**

1. If the authors have adequately addressed your comments raised in a previous round of review and you feel that this manuscript is now acceptable for publication, you may indicate that here to bypass the “Comments to the Author” section, enter your conflict of interest statement in the “Confidential to Editor” section, and submit your "Accept" recommendation.

Reviewer #1: All comments have been addressed

Reviewer #2: All comments have been addressed

2. Is the manuscript technically sound, and do the data support the conclusions?

Reviewer #1: Yes

Reviewer #2: Yes

3. Has the statistical analysis been performed appropriately and rigorously? 

Reviewer #1: Yes

Reviewer #2: Yes

4. Have the authors made all data underlying the findings in their manuscript fully available?

Reviewer #1: Yes

Reviewer #2: Yes

5. Is the manuscript presented in an intelligible fashion and written in standard English?

Reviewer #1: Yes

Reviewer #2: Yes

6. Review Comments to the Author

Reviewer #1: The research is relevant. The burnout syndrome is a disease that affects various professional categories and grow the number of research on this subject. In the Abstract the authors reported a high prevalence of burnout in French Neurosurgeons. However, in the final conclusion the authors did not make any reference to the burnout levels in this category of professionals. Also i did not realise the scale wich indicates the burnout levels are high or low.

Please, check it.

Reviewer #2: (No Response)

7. PLOS authors have the option to publish the peer review history of their article (what does this mean?). If published, this will include your full peer review and any attached files.

Reviewer #1: No

Reviewer #2: No

---

## [Author Response · Author response to Decision Letter 1]

18 Apr 2020

This is indeed a relevant point in the move of our paper that was missing. We compared our results with a meta-analysis of French physicians and American surgeons. Our conclusion of high prevalence of burnout was drawn on this comparison because such scale does not exist. Burnout definition and high, medium, or low level of each subscales (i.e. emotional exhaustion, depersonalization and personal accomplishment) were based on [1].

1. Maslach C, Jackson SE. The measurement of experienced burnout. J Organ Behav. 1981;2: 99–113. doi:10.1002/job.4030020205

Added text line 196: which is higher than the 29 % rate in comparison with the most comprehensive and complete meta-analysis of French physicians[17] following the same burnout definition. As for surgeons, Campbell et al[18] reported a 32% rate in a population of American surgeons.

Added text line 300: Burnout prevalence is high in comparison with physicians and even among surgeons.

---

## [Editor Report · Decision Letter 2]

29 Apr 2020

Personal and Psychosocial factors of Burnout: A Survey within the French Neurosurgical Community

PONE-D-20-00752R2

Dear Dr. Baumgarten,

We are pleased to inform you that your manuscript has been judged scientifically suitable for publication and will be formally accepted for publication once it complies with all outstanding technical requirements.

With kind regards,

Sergio A. Useche, Ph.D.

Academic Editor

PLOS ONE
---

## [Editor Report · Acceptance letter]

13 May 2020

PONE-D-20-00752R2 

Personal and Psychosocial factors of Burnout: A Survey within the French Neurosurgical Community 

Dear Dr. Baumgarten:

I am pleased to inform you that your manuscript has been deemed suitable for publication in PLOS ONE. Congratulations! Your manuscript is now with our production department. 

With kind regards,

on behalf of

Dr. Sergio A. Useche 

Academic Editor

PLOS ONE